



# River-enhanced non-linear overtide variations in river estuaries

Leicheng Guo[1, *], Chunyan Zhu [1, 2], Huayang Cai [3], Zheng Bing Wang [1, 2, 4], Ian Townend [5], Qing He [1]

[1] State Key Lab of Estuarine and Coastal Research, East China Normal University, Shanghai 200241, China

[2] Department of Hydraulic Engineering, Faculty of Civil Engineering and Geosciences, Delft University of Technology, Delft 2600GA, the Netherlands

[3] School of Marine Engineering and Technology, Sun Yat-Sen University, Guangzhou, China

[4] Marine and Coastal Systems Department, Deltares, Delft 2629 HV, the Netherlands

[5] School of Ocean and Earth Sciences, University of Southampton, Southampton, UK

* *Corresponding author*, E-mail: lcguo@sklec.ecnu.edu.cn

**Abstract:** Tidal waves traveling into estuaries are modulated in amplitude and shape due to bottom friction, funneling planform and river discharge. The role of river discharge on damping incident tides has been well-documented, whereas our understanding of the impact on overtide is incomplete. Inspired by findings from tidal data analysis, in this study we use a schematized estuary model to explore the variability of overtide under varying river discharge. Model results reveal significant $M_4$ overtide generated inside the estuary. Its absolute amplitude decreases and increases in the upper and lower parts of the estuary, respectively, with increasing river discharge. The total energy of the $M_4$ tide integrated throughout the estuary reaches a transitional maximum when the





river discharge to tidal mean discharge (R2T) ratio is close to unity. We further
identify that the quadratic bottom stress plays a dominant role in governing the
$M_4$ variations through strong river-tide interaction. River flow enhances the
effective bottom stress and dissipation of the principal tides, and reinforces
energy transfer from principal tide to overtide. The two-fold effects explain the
nonlinear $M_4$ variations and the intermediate maximum threshold. The model
results are consistent with data analysis in the Changjiang and Amazon River
estuaries and highlight distinctive tidal behaviors between upstream tidal rivers
and downstream tidal estuaries. The new findings inform study of compound
flooding risk, tidal asymmetry, and sediment transport in river estuaries.
**Key words**: River discharge; Overtide; Bottom stress; Estuary

**1. Introduction**
Tides are a primary force driving water motion and transport of sediment
and contaminant in estuarine and coastal environments. Examination of tidal
wave dynamics supports many aspects of coastal management, including
flooding risk mitigation, coastal erosion defense, and wetland conservation.
Tidal dynamics in oceanic and coastal waters have been extensively studied
for centuries (Green, 1837; Talke and Jay, 2020). It is already well established
that tidal waves traveling into estuaries are altered in amplitude and shape due
to water depth changes, channel convergence (Jay, 1991; Friedrichs and
Aubrey, 1994; Lanzoni and Seminara, 1998; Talke and Jay, 2020), and river
discharge (Godin, 1985; Horrevoets et al., 2004; Cai et al., 2014). Given tidal
wave celerity is a function of water depth in shallow environments, high water
travels faster than low water, leading to shorter rising tide and longer falling
tide, i.e., tidal wave deformation and tidal asymmetry. Tidal wave deformation
at the daily time scale is nicely represented by superimposition of $M_2$ and its
first overtide $M_4$ (Pugh, 1987). The amplitude of $M_4$ overtide is basically small
and insignificant in relatively deep and open coastal seas, but may become
profound inside tidal estuaries. The energy of $M_4$ overtide inside estuaries is



extracted from astronomical tides through the nonlinear processes (Parker,
1984; Talke and Jay, 2020). The behavior and dynamics of $M_4$ tide in
tide-dominant estuaries and lagoons have been extensively examined
because the resultant tidal asymmetry controls tide-averaged sediment
transport and morphological changes (Parker, 1984; Speer and Aubrey, 1985;
Friedrichs and Aubrey, 1988; Le Provost, 1991 etc.).
River flow enhances tidal energy dissipation and stimulates wave
deformation (Jay and Flinchem, 1997; Godin, 1999; Horrevoets et a., 2004;
Toffolon and Savenije, 2011). River discharge reinforces wave deformation by
prolonging falling tides and shortening rising tides, which is featured by larger
overtide amplitude under higher river discharge (Stronach and Murty, 1989;
Gallo and Vinzon, 2005). However, a small number of studies suggest that the
impact of river flow on tidal wave deformation and overtide generation exhibits
more spatial variability within river estuaries under varying river discharge. For
instance, Godin (1985, 1999) reported accelerated low water and retarded
high water in the upper Saint Lawrence Estuary under larger river discharge,
whereas the high water is hastened and the low water delayed in the lower part
of the estuary. In the Changjiang River estuary, the amplitude of the
quarter-diurnal tidal species (overtides and compound tides), resolved by
continuous wavelet transform method, becomes larger in the lower part of the
estuary, but smaller in the upper part of the estuary under high river flow
conditions (Guo et al., 2015). These findings imply that the $M_4$ overtide is
sensitive to river discharge magnitude and it displays different spatial
variations under different river discharge conditions, which is, however,
insufficiently understood.
The overtide generated locally within estuaries is inherently related to the
nonlinear dynamics in shallow waters. Non-linearity enters the mathematical
representation of a tidal system through the divergence of excess volume in
the continuity equation and the advection and bottom friction terms in the
momentum equation (Speer and Aubrey, 1985; Parker, 1984, 1991; Wang et al.





1999, 2002). Pioneering studies with scaling analysis suggested that the
advection term is insignificant when scaled with estuarine length or wavelength
in short and tide-dominated estuaries, thus was ignored in past analytical study
of tides (Speer and Aubrey, 1985; Friedrichs and Aubrey, 1994). However, in
the presence of a river flow, the advection term may play a role in slowing
down incident tidal waves and speeding up the reflected waves (Godin, 1985,
1991; van Rijn, 2011; Kästner et al., 2019). This is because river flow enlarges
the mean current, therefore the advection term becomes significant and cannot
be ignored in river estuaries (Talke and Jay, 2020). Parker (1984) provided a
thorough analysis of the importance of frictional effects on tidal interactions.
The quadratic bottom shear stress has the effect of reducing tidal amplitudes
and decreasing wave celerity (Proudman, 1953; Godin, 1985, 1991, 1999; Jay,
1991; Horrevoets et al., 2004), and stimulating the generation of new
harmonics (Proudman, 1953; Pingree and Maddock, 1978; Parker, 1984, 1991;
Wang et al., 1999). Given all the three nonlinear terms are attributed to
creating forced harmonics (Parker, 1984; Walters and Werner, 1991; Wang et
al., 1999), it would be helpful to determine their relative importance. Gallo and
Vinzon (2005) provided an evaluation of the relative importance of the
nonlinear terms on overtide for the Amazon River estuary. But the results were
only presented for a mean river discharge condition. It still remains an open
question as to which nonlinear term plays a more significant role under varying
river discharge conditions.

In this contribution we deploy a numerical model to explore river-tide

interaction and subsequent impact on overtide behavior in a schematized long
estuary. We aim to explore 1) how varying river discharge would modulate the
overtides, and 2) what is the controlling impact of the nonlinear terms on the
spatial variability of overtide under different river discharge.

**2. Model setup and data analysis**
**2.1 Inspirations from the Changjiang River estuary**





The rationale of this study comes from tidal analysis in the Changjiang
River estuary, which is a large tidal system with a tide-influenced river reach as
along as 650 km (Figure 1a). River discharge at the tidal wave limit, Datong,
varies seasonally in the range of 10,000-60,000 $m^3$/s in the post-Three Gorges
Dam period (Figure 1b; Guo et al., 2018). The incoming tides are semi-diurnal
with a maximum tidal range of 5.9 m, and the $M_2$ is the most significant
constituent, followed by $S_2$, $O_1$, and $K_1$. Based on harmonic analysis
(Pawlowicz et al., 2002) of tidal height data in the time periods when river
discharge varies in a small range (close to a stationary situation, see Guo et al.
(2016) for further details), we see that the incoming tidal waves are firstly
amplified before they travel into the estuary, owing to a landward decrease in
water depth (Figure 1c). They are, however, predominantly dissipated inside
the estuary, despite the width convergence in the lower part of the estuary
seaward of Jiangyin, because of stronger influence of bottom friction and/or
river discharge. The river-enhanced tidal damping is more significant in the wet
season when the river discharge is higher, particularly in the upper part of the
estuary upstream of Jiangyin.

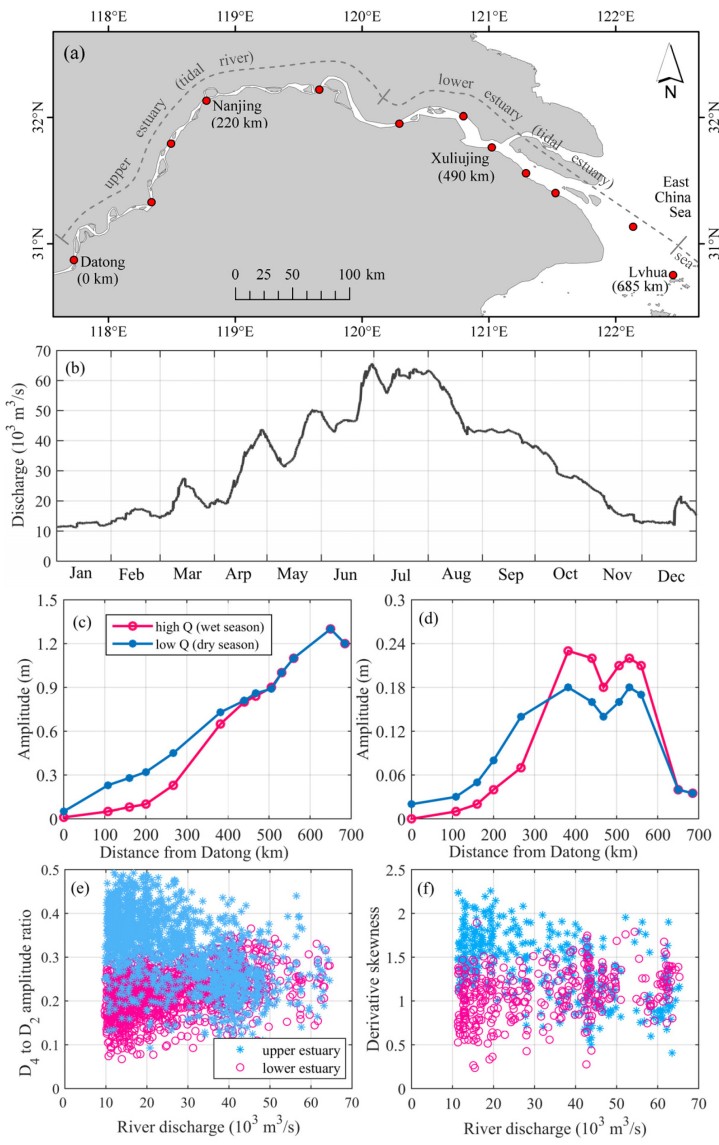

**Figure 1.** (a) The geometry and tidal gauges in the Changjiang River Estuary,

(b) river discharge variations within a year course, along-river (c) $M_2$, and (d)

$M_4$ amplitude variations in the dry and wet seasons, (e) amplitude ratios of the

quarter-diurnal to semi-diurnal tides, and (f) skewness of the time derivative of

tidal water levels in the upper (Nanjing) and lower (Xuliujing) estuaries. Details

of the Changjiang River estuary and the tidal data are given in Guo et al.

(2015). The numbers in the brackets in panel (a) indicate the seaward distance


from Datong. The data in panels (c)-(e) is from Guo et al. (2016) and that in
panel (f) is from Guo et al. (2019).

A significant $M_4$ overtide is detected inside the estuary while it is
insignificant to seaward of the estuary (Figure 1d). The smaller $M_4$ amplitude in
the region km-380 and km-520 in both dry and wet seasons is attributed to
interaction between the two main branches around Xuliujing. Apart from that,
the $M_4$ amplitude is larger in the lower part of the estuary in the wet season
when the river discharge is higher (Figure 1d). Moreover, the amplitude ratios
of the quarter- to semi-diurnal tidal species (derived by continuous wavelet
transform) decrease with increasing river discharge in the upper part of the
estuary but increase in the lower estuary (Figure 1e; Guo et al., 2015). Similar
analyses, using the skewness of the time derivative of tidal water levels, show
that the duration asymmetry between falling and rising tides exhibits similar
variations as the amplitude ratios (Figure 1f; Guo et al., 2019). These results
regarding the longitudinal $M_4$ amplitude variations by harmonic analysis
(Figure 1d), the amplitude ratios of the quarter- to semi-diurnal tidal species
(Figure 1e), and the derivative skewness variations (Figure 1f) consistently
demonstrate that the overtides display distinctive variations between the upper
and lower parts of the estuary in response to low and high river discharge
conditions. However, such changing behavior was insufficiently discussed in
previous studies. One challenge is that the river discharge varies continuously
in reality and induces non-stationary variations in tidal dynamics. Conventional
harmonic analysis is unable to accurately resolve the tidal changes when the
river discharge varies continuously in a big range (Jay and Flinchem, 1997;
Jay et al., 2014; Guo et al., 2015). Specifically, it is unknown how the overtides
will behave as the river discharge varies between the low and high limits other
than the results shown in Figure 1d.

**2.2 Model setup**





Examination of tidal data has provided a basic framework for our
understanding of tidal dynamics (Dronkers, 1964; Godin, 1985). However,
conventional harmonic analysis may not accurately resolve tidal constituents
owing to the non-stationary river discharge variations (Jay and Flinchem,
1997). In addition, analytical solutions of the tidal dynamic equations have
facilitated exanimation of leading-order wave propagation such as landward
damping or amplification of astronomical tides, given its advantages in terms of
fast setup and transparency in unraveling physical processes (Jay, 1991;
Friedrichs and Aubrey, 1994; Lanzoni and Seminara, 1998; Savenije, 2005).
However, analytical models usually assume tidal propagation as a single wave
component, based on simplified tidal dynamic equations after scaling analyses,
e.g., adopting a linear assumption or a nonlinear expansion of the friction term
(Green, 1837; Kreiss, 1957; Jay, 1991; Parker, 1991; Friedrichs and Aubrey,
1994; van Rijn, 2011). Analytical models may not fully capture the nonlinearity
imbedded in tidal dynamics, considering that the importance of different
nonlinear terms is likely not the same in different parts of long systems,
particularly under strong river flow conditions in long estuaries.
In this study we seek to capture the nonlinear dynamics by using a
numerical model, i.e., the open-source Delft3D codes, which has been widely
validated and used in varying estuarine and coastal environments (Lesser et
al., 2004). We construct a schematized 1D estuary model with a convergent
planform mimicking the Changjiang River estuary. The model domain
describes a 650 km long estuary that is composed of a weakly convergent
upstream segment (km-0 to km-400, width varying from 2 to 5 km) and a
strongly convergent downstream segment (km-400 to km-650, width varying
from 5 to 32 km (Figure 2a). Another situation with a uniform prismatic channel
(i.e., 2 km width and similar length) is adopted as part of the sensitivity analysis
to see the influence of basin geometry (Figure 2b).



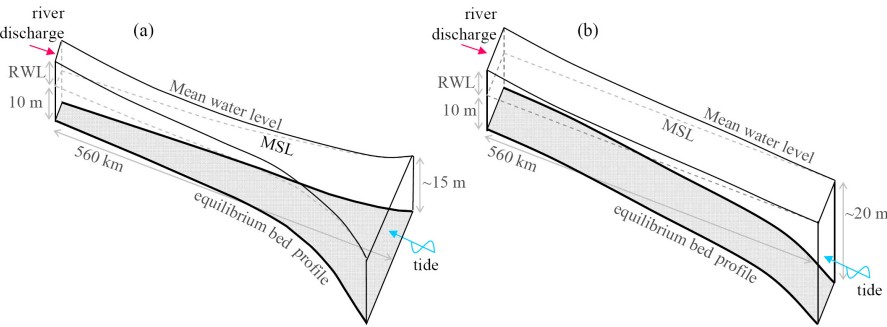

**Figure 2**. Sketches of the schematized estuary model outline and settings considering (a) a convergent and (b) a prismatic planform. The shade face indicates the equilibrium bed profile. The RWL and MSL indicate residual water level and mean sea level, respectively.

The model is forced by river discharge and tides. A combination of different tidal constituents is imposed, and for simplicity we mainly consider a semi-diurnal $M_2$ constituent with an amplitude of 1.0 m. Extra simulations considering both $M_2$ and $S_2$ constituents (an amplitude of 0.5 m) are included to facilitate more tidal interactions and generation of representative compound tide such as $MS_4$. Other astronomical constitutes like $O_1$ and $K_1$ are excluded because they would not affect the $M_2$ propagation very much. River discharge is prescribed by constant values of 0, 10,000, 30,000, 60,000 $m^3$/s, symbolized as Q0, Q1, Q3, and Q6 scenarios, respectively, to facilitate harmonic analysis with a stationary assumption. A dimensionless parameter, defined as the ratio of river discharge to tide-averaged mean discharge (i.e., tidal prism divided by tidal period) at the mouth section (R2T ratio), is estimated to be 0, 0.5, 2.6, and 42, which can be classified into tide-dominant, low, medium, and very high river discharge circumstances, respectively (see section 3.3). The size of the schematized estuary and the forcing conditions are characterized for a large river estuary and in this case key dimensions from the Changjiang River Estuary are used.

To obtain a suitable bottom profile for the tidal model, we first run a





morphodynamic simulation based on the above-mentioned model outline, with
an $M_2$ tide and a river discharge seasonally varying between 10,000 and
60,000 $m^3$/s as the boundary forcing conditions, as that in Guo et al. (2016).
The long-term morphodynamic simulation starts from an initial sloping bed with
depth varying from 5 m to 15 m seaward, considers sediment transport and
bed level changes, which leads to a morphodynamic equilibrium when bed
level changes become small at the time scales greater than decades (Guo et
al., 2016). The eventual equilibrium bed profile is then used as the bottom level
condition in the tidal simulations. The purpose of using this equilibrium bed
profile is to maintain consistency between the forcing and morphological
conditions. Based on this equilibrium bed profile and given high river discharge
imposed, the incoming tides are largely dissipated in the landward region of
the estuary, thus the influence of wave reflection is minimized. Details of the
morphodynamic model can be found in Guo et al. (2016), thus are not
repeated here.
Past studies using similar 1D representation of tidal estuaries confirm the
capture of leading-order dynamic processes (Friedrichs and Aubrey, 1994;
Lanzoni and Seminara, 1998). But it is noteworthy that the 1D model excludes
tidal flats and assumes uniform water density. These excluded processes may
have additional impact on tidal dynamics, e.g., additional momentum loss,
reduction in bottom drag, and extra tidal asymmetry (Friedrichs and Aubrey,
1988; Talke and Jay, 2020). Although simplified, the model provides a virtual
lab where tidal wave propagation, deformation and associated overtide
dynamics under varying river discharge can be isolated from the influences of
basin geometry and irregular shoreline, which enables straightforward
exploration of river-tide interactions.

**2.3 Data analysis**
The 1D model solves the width-averaged shallow water equations, i.e., the
continuity and momentum conservation equations, when the effect of Coriolis

force and density variations are neglected (Dronkers, 1964), as follows,

$$\frac{\partial \eta}{\partial t} + \frac{\partial u(h+\eta)}{\partial x} = 0 \tag{1}$$

$$\frac{\partial u}{\partial t} + u\frac{\partial u}{\partial x} + g\frac{\partial \eta}{\partial x} + \frac{gu|u|}{C^2(h+\eta)} = 0 \tag{2}$$

where $u$ is velocity, $\eta$ is water height above mean sea level, $h$ is water depth below mean sea level, $g$ is gravitational acceleration (9.8 m$^2$/s), and $C$ is a Cheźy friction coefficient prescribed as 65 m$^{1/2}$/s uniformly.

As the bed level is prescribed as an equilibrium profile, the water depth $h$ is constant. In the presence of a river discharge, the water level height is composed of two parts, namely a mean water height related to river flow $\eta_0$, and a tide-induced water level oscillation,

$$\eta(x,t) = \eta_0(x) + \eta_{M2}(x)\cos(\omega t - kx) \tag{3}$$

in case of the presence of M$_2$ tide only, in which $\eta_{M2}$ is the surface amplitude of M$_2$, and $\omega$ is the frequency of M$_2$, and $k$ is tidal wave number. Similarly, the current is composed of a mean current and a tidal component,

$$u(x,t) = u_0(x) + u_{M2}(x)\cos(\omega t - kx - \theta) \tag{4}$$

in which $u_0$ is the mean current velocity, $u_{M2}$ is the velocity amplitude of M$_2$, $\theta$ is the phase difference between tidal surface wave and tidal currents.

Three nonlinear terms are identified in the tidal wave equations, namely the discharge gradient term in the continuity equation, and the advection and quadratic friction terms in the momentum equation:

● discharge gradient: $\dfrac{\partial u(h+\eta)}{\partial x} = \dfrac{\partial(uh)}{\partial x} + \dfrac{\partial(u\eta)}{\partial x}$ $\tag{5}$

● advection: $u\dfrac{\partial u}{\partial x} = \dfrac{\partial}{\partial x}(\dfrac{u^2}{2})$ $\tag{6}$

● bottom friction: $\dfrac{gu|u|}{C^2(h+\eta)} \approx \dfrac{g}{C^2}(\dfrac{u|u|}{h} - \dfrac{\eta u|u|}{h^2})$ $\tag{7}$

The bottom friction term is approximately expanded into a bottom shear stress term and a term considering depth variations, as the two terms on the right hand of Eq. (7), respectively, according to Godin and Martinez (1994), given the tidal amplitude to water depth ratio ($|\eta|/h$) is generally smaller than



one. Note that the bottom friction term can be calculated accurately with
resolved water depths and velocities in the numerical model, while the
approximation of Eq. (7) is just used to analytically demonstrate how the
friction would lead to local generation of compound tides and overtides. Firstly
considering a situation when river discharge is small and the associated mean
current ($u_0$) is insignificant, the quadratic bottom stress can be further
expressed by Fourier decomposition according to Le Provost (1991) and Wang
et al. (1999):
$$\frac{u\,|u|}{h} \approx \frac{u_{M2}^2}{h}\sum_{n=0,1,2,\ldots}(-1)^{n+1}\frac{8}{(2n-1)(2n+1)(2n+3)\pi}\cos(2n+1)\omega t\cos(nkx) \qquad (8)$$

Equation (8) suggests that the self-interaction of $M_2$ tide through the
quadratic bottom stress produces a series of overtide harmonics with
odd-multiple frequencies, e.g., $M_6$ and $M_{10}$ (Parker, 1984). In addition, Eq. 8
also yields a contribution to the same frequency as $M_2$ (when n=0), which
suggests tidal energy dissipation via the quadratic shear stress term (Wang et
al., 1999). Similarly, the depth variation term in Eq. 7 can be expressed as:
$$\frac{\eta u\,|u|}{h^2} \approx \frac{\eta_{M2}u_{M2}^2}{h^2}\sum_{n=0,1,2,\ldots}(\ldots)\ \cos(\omega t)\cos[(2n+1)\omega t]$$

$$= \frac{\eta_{M2}u_{M2}^2}{h^2}\sum_{n=0,1,2,\ldots}(\ldots)\ [\frac{1}{2}\cos(2n\omega t)+\frac{1}{2}\cos(2n+2)\omega t] \qquad (9)$$

Equation (9) suggests that the self-interaction of $M_2$ tide through the depth
variation term generates even-multiple frequency harmonics, e.g., $M_4$ and $M_8$.
Similar decomposition analysis for the advection and discharge gradient term
suggests the generation of even-frequency overtide as well (Parker, 1984;
Wang et al., 1999). Following similar logic, when two components such as $M_2$
and $S_2$ tides are prescribed, compound tides with frequencies the sums (e.g.,
$MS_4$) or differences (e.g., MSf) of the prescribed constituents are generated.
The main focus of this study is devoted to $M_4$ overtide, given it is the first
overtide of $M_2$ and of profound importance for tidal asymmetry.
Following the above analyses that qualitatively indicates the possibility of





local overtide generation, we attempt to further quantify the relative
contribution of the nonlinear terms. For this purpose, we employ another
approximation of the quadratic shear stress term, as follows, according to
Godin and Martinez (1994) and Godin (1999),

$$u \mid u \mid \approx 0.35u + 0.71u^3 \tag{10}$$

Replacing the Eq. (7) with Eq. (10) and using the expansion of Eqs. (3) and
(4), a harmonic decomposition using the sine and cosine summation rules is
used to identify the contribution of the nonlinear discharge gradient, advection,
and bottom friction terms based on Eqs. (5) to (7). It follows the method in
Gallo and Vinzon (2005) and Leberthal et al. (2019), but considering both
quadratic bed shear and depth variation terms.
• discharge gradient: $\quad 0.5u_{M2}\dfrac{\mathrm{d}\eta_{M2}}{\mathrm{d}x} + 0.5\eta_{M2}\dfrac{du_{M2}}{\mathrm{d}x}$ \hfill (11)
• advection: $\quad 0.5u_{M2}\dfrac{du_{M2}}{\mathrm{d}x}$ \hfill (12)
• friction: $\quad \dfrac{1.065g}{C^2h}u_0u_{M2}^2 +$ \hfill (13)

$$\dfrac{g}{C^2h^2}[1.065\eta_0u_0u_{M2}^2 + 0.525u_0^2\eta_{M2}u_{M2} + 0.355\eta_{M2}u_{M2}^3]$$

The first term in Eq. 13 is ascribed to the quadratic bottom shear while the
other terms are attributed to the depth variations. Again, the Eqs. (11) to (13)
suggest that the interaction between the mean current and $M_2$ velocity would
generate even-frequency harmonics like $M_4$ via both the quadratic bed shear
stress and depth variation terms, implying river influence through river-tide
interaction. Harmonic analyses of the model-output time series of water levels
and currents provide mean water height, mean current, and the amplitudes
and phases of surface wave and velocity of $M_2$ and $M_4$ tides for Eqs. (11) to
(13). To indicate their relative importance, the advection and friction terms are
then normalized by squared maximum velocity, and the discharge gradient
term is normalized by the product of maximum velocity and maximum water
level range. Comparison of the four scenarios forced by different river
discharge thus helps to demonstrate the variability (see section 3.3).






### 3. Model results

### 3.1 Tidal variations under varying river discharge

The longitudinal amplitude variations of both the principal and forced
constituents are shown in Figure 3. The $M_2$ tide is firstly slightly amplified in the
utmost seaward regions close to the mouth, owing to channel convergence,
(Figure 3a). Landward of that, the incoming $M_2$ tide is predominantly dissipated,
and river discharge enhances the damping. In addition, a considerable $M_4$ tide
is detected in the Q0 scenario (no river discharge) with a local amplitude
maximum around km-450. The $M_4$ amplitude becomes larger throughout the
estuary in the Q1 scenario compared with that that in Q0 (Figure 3b). However,
under higher river discharge, the $M_4$ amplitude reduces in the upper part of the
estuary, e.g., landward km-300, but continues to increase in the lower reaches,
e.g., seaward km-500 (Figure 3b). The location with maximal $M_4$ amplitude
moves slightly landward as the river discharge increases from zero (i.e., from
km-450 in the Q0 scenario to km-400 in the Q1 scenario), but seaward as the
river discharge further increases (i.e., from km-420 in the Q3 scenario to
km-500 in the Q6 scenario). The $M_4$ to $M_2$ amplitude ratio exhibits similar
variations as the absolute $M_4$ amplitude, but the ratio is overall larger in the
upper parts of the estuary in which the absolute amplitudes of both $M_2$ and $M_4$
tides are small (Figure 3d). The increasingly damped and distorted tidal waves
further illustrate the river impact on the incoming tides (see Figure S1 in the SI).
When both $M_2$ and $S_2$ are imposed in the seaward boundary, a compound
overtide $MS_4$ is detected inside the estuary, which exhibits similar spatial
variations as the $M_4$ tide (Figure 3c). We then focus on the $M_4$ overtide in the
following discussions.



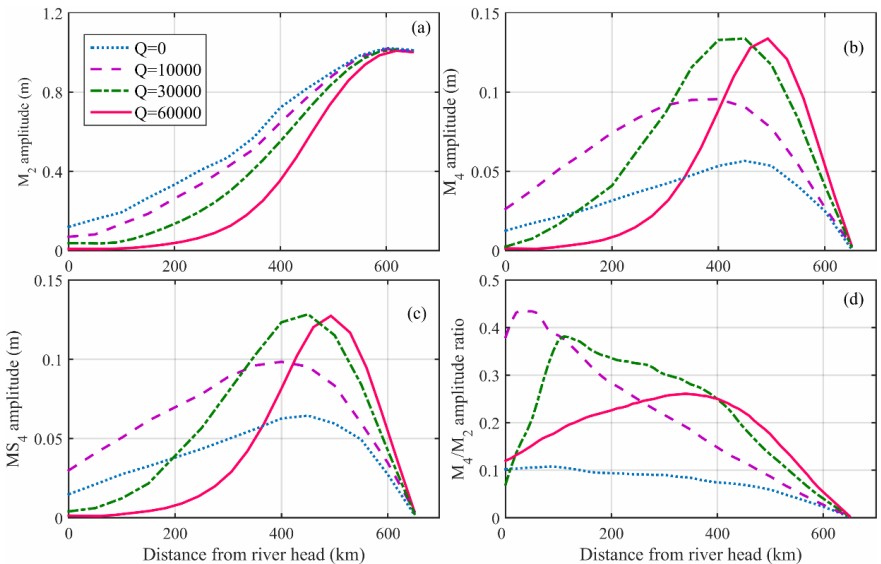

**Figure 3**. The model-reproduced longitudinal variations of (a) $M_2$ tidal amplitude, (b) $M_4$ tidal amplitude, (c) $MS_4$ amplitude (in the extra scenario when both $M_2$ and $S_2$ are imposed at the boundary), and (d) the $M_4$ to $M_2$ amplitude ratios in the convergent estuary.

**3.2 Sensitivity to channel convergence**

Channel convergence is expected to affect tidal wave propagation and wave deformation (Jay, 1991; van Rijn, 2011; Talke and Jay, 2020). To demonstrate the sensitivity of the model results to width variations, we setup a prismatic channel model with similar settings as the convergent estuary. A close-to-equilibrium bed profile is again obtained via morphodynamic simulation for the prismatic estuary. River discharge elevates the mean water level and mean current (Figure 4a). The incoming $M_2$ tide is overall damped inside the estuary, without any amplification (Figure 4b). Similar $M_4$ overtide is generated as well, but its amplitude is approximately 30% smaller than that in the convergent estuary (Figure 4c). A smaller tidal prism owing to a smaller mouth width and surface area explain the smaller tidal amplitude in the rectangular estuary. Apart from the differences in the absolute amplitudes, the



longitudinal variations of both the principal and forced tides and their spatial
dependence on river discharge exhibit similar patterns as the convergent
estuary (Figures 3 and 4). These consistent results imply that channel
convergence does not fundamentally changes the spatial dependence of
overtide behavior on river discharge, thus the findings from the prismatic
estuary is taken to have implications for estuaries in general and will be the
focus of further more detailed examination in order to highlight the controlling
impact of river discharge.

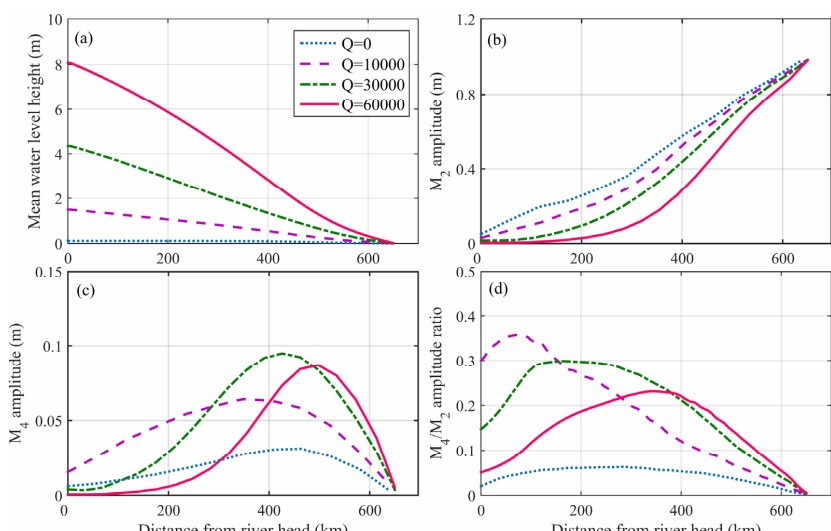


**Figure 4**. Model-reproduced longitudinal variations of (a) mean water level
height, (b) M$_2$ tidal amplitude, (d) M$_4$ tidal amplitude, and (d) the M$_4$ to M$_2$
amplitude ratio under different river discharge in the prismatic estuary.

**3.3 Contribution of the nonlinear terms**
We then use the harmonic decomposition method proposed in section 2.3
to quantify the contribution of the different nonlinear terms on M$_4$ variations. In
the absence of river discharge (Q0 scenario), the discharge gradient term is
the largest contribution owing to strong landward damping of M$_2$ and
subsequent longitudinal flux gradients, followed by bottom friction and
advection (Figure 5a). The bottom friction term becomes more significant in the



presence of river discharge (Figures 5b-d). The impact of the quadratic bottom
shear stress is much more important than that of depth variations. The
influence of the advection term is relatively small compared to the other two
nonlinear terms. Spatially, the impact of bottom friction is more profound in the
upper parts of the estuary, whereas the impact of discharge gradient and
advection is more apparent in the regions close the estuary mouth. The
location of maximal $M_4$ amplitude coincides with the peak in the combined
contribution of discharge gradient and advection in the Q0 scenario, but with
the peak in bottom friction in the other three scenarios. Overall, these results
suggest that the three nonlinear terms are equally important in the
tide-dominated long estuary, whereas the importance of the bottom friction, or
more precisely the quadratic bottom shear stress, stand out when there is
significant river discharge.

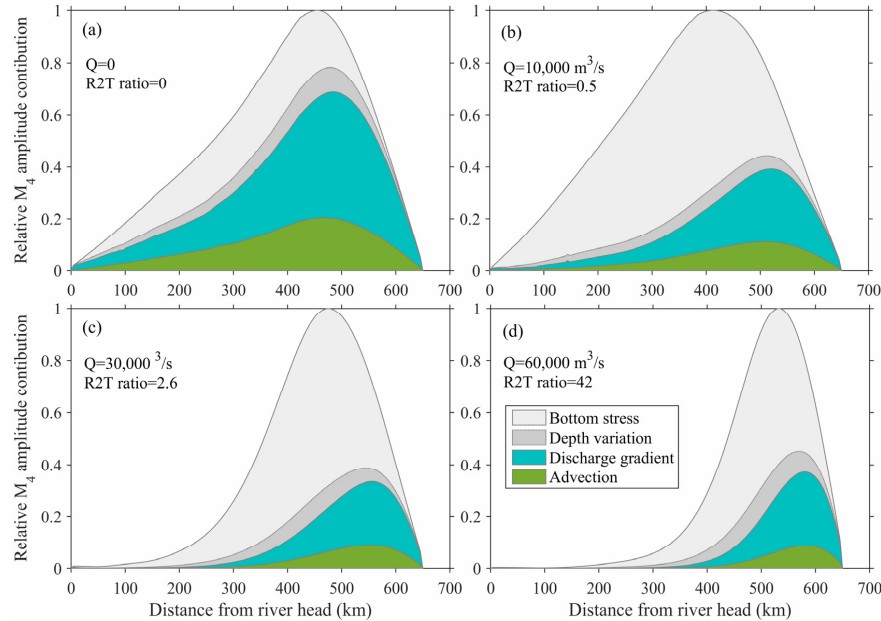


**Figure 5**. Quantification of the relative importance of three nonlinear terms on

$M_4$ overtide amplitude in the (a) Q=0 (R2T ratio=0), (b) Q=10,000 (R2T

ratio=0.5), (c) Q=30,000 (R2T ratio=2.6), and (d) Q=60,000 $m^3$/s (R2T ratio=42)

scenarios. The contribution of bottom friction is divided into the components of



bottom shear stress and depth variation. The relative $M_4$ amplitude is
normalized by the maximal value in each scenario.

The importance of the quadratic bottom shear stress can be furthermore
inferred when comparing the model results under quadratic and linear bottom
shear stress. The quadratic bottom shear stress can be linearized using the
first order of the energy dissipation condition (Lorentz, 1926), as that applied
by Zimmerman (1992) and Hibma et al. (2003) (see SI for more details). When
similar simulations are run using linear bottom stress, the landward damping
rates of the principal tides become smaller (see Figure S4). Measurable $M_4$
overtide is still detected, which is ascribed to the effects of other nonlinear
effects (e.g., the advection and depth variations), but its amplitude is
comparably smaller than that under a quadratic bottom stress (Figure S4).
Moreover, increasing river discharge neither induces more tidal wave damping,
nor more overtide generation under a linear bottom stress.

**3.4 Quantification of the river discharge threshold**
The abovementioned model results imply that the $M_4$ amplitude tends to
first increase and then decrease as the principle $M_2$ is increasingly dissipated
by larger river discharge. To better reveal the nonlinear variations, we run extra
simulations under constant river discharge in the range of 0 to 60,000 $m^3$/s at
an increment of 5,000 $m^3$/s. Since the tidal amplitudes vary along the estuary,
we then integrate the total (tide-averaged) energy of $M_2$ and $M_4$ tides (kg·$m^2$/$s^2$)
throughout the estuary to represent overall tidal strength (van Rijn, 2011) by:
$$\int_0^L 0.5\rho g b A(x)^2 \, dx / L \qquad (14)$$

where: $L$ is the channel length, $\rho$ is the water density, $b$ is channel width which
is uniform in the rectangular channel, and $A$ is the surface amplitude of the $M_2$
or $M_4$ tide which varies along the estuary.
We see that the total energy of the $M_2$ tide decreases approximately



exponentially with increasing R2T ratios (Figure 6a). The decrease is more
significant for R2T<5 (see Figure S2a). The total energy of $M_4$ overtide,
however, first increases with increasing R2T ratio from zero and reaches a
peak when the R2T ratio is approximately unity, followed by a decrease as the
R2T ratio further increases (Figure 6a). The maximum total energy of $M_4$ is 4.7
times the case with no river discharge, under the model framework in this study.
Similarly, the energy ratio of $M_4$ to $M_2$ displays similar variations as the total
energy variation of the $M_4$ tide, with a peak reached when the R2T ratio is
around 1-2 (Figure 6b). These results confirm that an intermediate river
discharge with R2T ratio close to unity, benefits maximal $M_4$ overtide
generation. Below this threshold, increased river discharge favors more $M_4$ tide
generation, whereas a larger river discharge above the threshold constrains
$M_4$ generation.

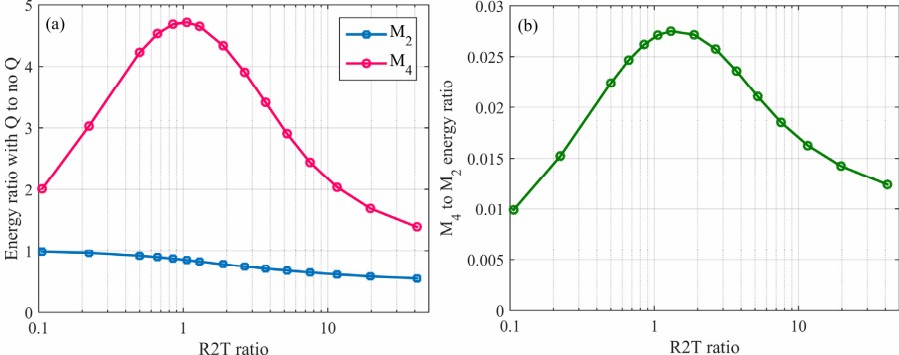


**Figure 6**. (a) The ratio of the total energy of $M_2$ and $M_4$ tides integrated
throughout estuary in the scenarios with river discharge to the case without
river discharge, and (b) the total energy ratio of $M_4$ tide to $M_2$ tide, as a function
of the ratio of river discharge to tide-mean discharge at the estuary mouth
(R2T ratio). Also see Figure S2 in the SI.

**4. Discussion**
**4.1 Comparison with actual estuaries**
The findings regarding overtide variability in the model are overall





consistent with data analyses in actual estuaries like the Changjiang and
Amazon River estuaries. The modeled results between the Q1 and Q3
scenarios are consistent with the along-channel variations of the principle tide
and overtide under low and high river discharge in the Changjiang River
Estuary. The changes in the $M_4$ to $M_2$ amplitude ratios with varying river
discharge between the upper and lower parts of the Changjiang River Estuary
also agree well with model results in the schematized convergent estuary (see
Figure S3). In the Amazon River estuary where the river discharge is similarly
high and varies in a large range, the $M_4$ amplitude was overall larger under a
mean river discharge throughout the estuary compared to an idealized
situation with zero discharge (Figure 7c; Gallo and Vinzon, 2005). This result is
qualitatively consistent with the modeled differences between the Q0 and Q1
scenarios in this study. In the Columbia River estuary, a maximum in $M_4$
amplitude is approached in the lower part of the estuary, followed by a
subsequent decrease upriver under a year-mean river discharge (Figure 7d;
Jay et al., 2014). The model results can also explain why a higher river
discharge hastened the high water and delayed the low water in the lower part
of the Saint Lawrence Estuary (Godin, 1985, 1999). These field data and
model results confirm that the findings regarding the spatial dependence of
overtide on river discharge are likely to be ubiquitous for river estuaries.

Similar reports of the overtide behavior were, however, not widely found in

many other estuaries, given that the tidal dynamics have been intensively
studied worldwidely. We think that it maybe because the majority of estuaries
worldwide are tide-dominated, such that river discharge is overall small and
rarely reaches a magnitude that exceeds R2T=1. Therefore, the role of river
discharge in stimulating wave deformation and associated overtide generation
has been widely observed and confirmed (when R2T<1), whereas the further
changes when R2T>1 are far less prevalent and hence less well documented.
Another possible explanation is that most tidal estuaries are small in physical
length (compared to wavelength), hence the spatial variations are less



apparent compared to long estuaries such as the Amazon, Changjiang, and
Columbia systems.

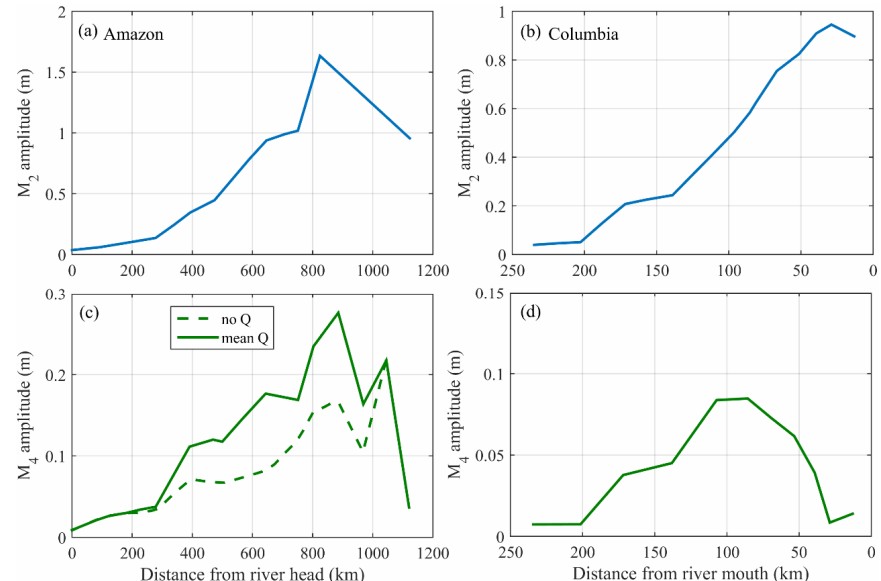


**Figure 7**. Amplitude variations of M2 and M4 overtide in the (a, c) Amazon
River estuary and (b, d) Columbia River estuary, from Gallo and Vinzon (2005)
and Jay et al. (2014), respectively.

**4.2 Role of river discharge**

The majority of past studies have focused on either the damping effect of

river flow on principal tides or the enhancing effect on overtide. When linking
them together, we see that the two-fold effects of river flow nicely explain the
nonlinear overtide changes in river estuaries (Figure 8). River discharge
enlarges the currents and the effective friction on the moving flow. It induces
more damping of the principal tides, i.e., more energy dissipation of incident
tides. In addition, river-enhanced bottom friction reinforces the energy transfer
from the principal tide to overtide, i.e., stimulated overtide generation. As more
principal tidal energy is dissipated, particularly in the landward part of estuaries,
the energy available for transfer to overtide is also constrained. As a result, an
intermediate river discharge (when the R2T ratio is close to unity) provides an
effective bottom stress that will not dissipate the principal tides too much, and
at the same time stimulates considerable energy transfer to overtides, leading
to the occurrence of a maximum in overtide energy. Note that other
high-frequency overtides display similar spatial variations as $M_4$, e.g., $MS_4$ (see
Figure 3c) and $MN_4$ tide when $M_2$ and $N_2$ constituents are prescribed (not
shown).
River impact on tidal wave propagation is space-dependent. River
discharge substantially elevates the mean water level in the upper part of
estuaries; the consequent larger water level gradient restricts landward wave
propagation (Godin, 1985; Cai et al., 2019). In the lower part of estuaries
where the incident tides are less dissipated, river flow plays a more important
role in reinforcing the effective bottom friction. As a result, dissipation of the
principal tide is more prominent in the upper part of estuaries, while tidal
energy transfer and overtide generation is more substantial in the lower part of
estuaries (Figures 8). These space-dependent dynamics explain the
contrasting behavior of overtide in response to increasing river discharge, and
also confirm that tidal wave deformation maybe one of the degrees-of-freedom
of estuaries to maintain a state of minimum work by adjusting tidal wave
shapes in response to different river discharge (Zhang et al., 2016).

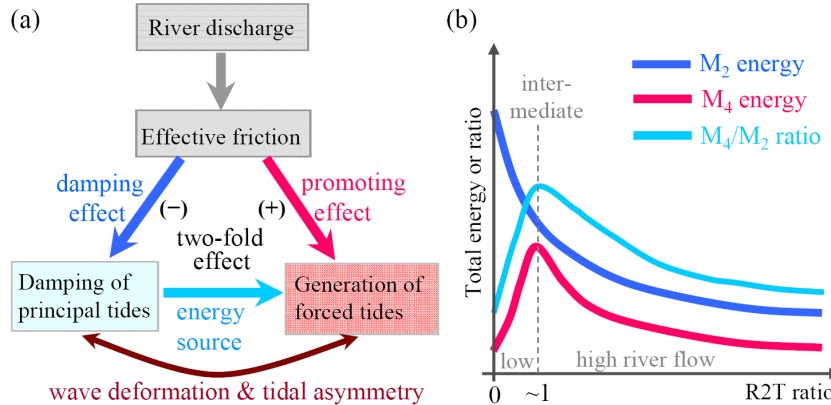


**Figure 8.** Sketches (a) showing the two-fold effects of river discharge on tides
mainly through the bottom friction, and (b) showing the intermediate river
discharge threshold that benefits maximum overtide generation.

The two-fold river impact on tidal propagation is coherently related to the
bottom friction. Past studies have indicated that the effects of river flow on tidal
damping are exerted by a mechanism identical to bottom stress (Horrevoets et
al., 2004; Cai et al., 2014). River-tide interaction enhances the bottom stress,
which subsequently induces larger tidal damping (Alebregtse and de Swart,
2016). Past studies have also suggested that the nonlinear advection term is
the main cause of $M_4$ generation in tide-dominant estuaries, whilst the
nonlinear bottom stress term leads to generation of $M_6$ (Pingree and Maddock,
1978; Parker, 1984, 1991; Wang et al., 1999). In this work we see that the
quadratic bottom stress term also leads to significant $M_4$, through river-tide
interaction, i.e., between a river-enhanced mean current and $M_2$ current. This
explains why the $M_4$ amplitude is larger in the presence of a river discharge
and a quadratic bottom stress, compared to the situation with no river
discharge and/or a linear bottom stress.
Given the comprehensive past studies of tidal dynamics in estuaries, the
contribution of this work lies in revealing the nonlinear overtide changes and
identification of a river discharge threshold that benefit maximum overtide
generation. A river flow above or below the threshold induce contrasting
overtide behavior along estuaries. Although the model results are obtained
under constant river discharge, the findings still hold true when considering
time-varying river discharge (see SI). One slight difference is that the tidal
damping rate would be slightly different during the rising and falling limb of a
river discharge hydrograph (Sassi and Hoitink, 2013), which may be due to a
time lag in the influence of river discharge along the length of the estuary.

**4.3 Implications and limitations**
Better understanding of the overtide behavior has implication for studies of
tidal bores, interpretation of extreme high water and associated flood risk, and


tide-averaged sediment transport. Tidal wave deformation changes the height
of high water and low water, which may then influence flooding risk
management and the water depth of navigational channels. Tidal bores are an
extreme phenomenon of tidal wave deformation when tides are concurrently
amplified and distorted to some degree (Bonneton et al., 2015). Tidal bores are
less likely to occur in river estuaries because of river-enhanced damping,
although deformation is enhanced. The interaction between tidally-averaged
mean current and quarter-diurnal overtide current may contribute to net water
transport (Alebregtse and de Swart, 2016). Tide-averaged sediment transport
induced by tidal asymmetry related to $M_2$-$M_4$ interaction plays a profound role
in controlling sediment import or export and resultant infilling or empty of
estuaries (Postma, 1961; Guo et al., 2014). It is noteworthy that the horizontal
velocity of the quarter-diurnal tide may exhibit more spatial variations than its
surface amplitude, owing to interaction with estuarine morphology and
inter-tidal interactions of eddy viscosity etc. (Dijkstra et al., 2017b; Lieberthal et
al., 2019).

Although we have argued that channel convergence will not fundamentally

change the model results and main findings, the potential impact of the
simplified model setting in this study still mandates careful evaluation when
applying them to actual estuaries. For instance, regional narrowing and
shallowness in geometry and morphology is expected to induce variations in
tidal damping rates and distribution of amplitudes. River-influenced estuaries
can be partially or highly stratified, and a density difference and associated
stratification affect tides by reducing the effective drag coefficient and changing
the pressure-gradient term (Talke and Jay, 2020). This impact maybe further
manifested in surface amplitude of overtide given the role of river-tide current
interaction in the nonlinear terms (Dijkstra et al. 2017a). Inter-tidal flats are
known as a sink of momentum and would exert additional impact on tidal wave
propagation (Hepkema et al., 2018). Exclusion of inter-tidal flats in this work
thus may lead to overestimation of the overtide amplitude. Furthermore, the



intermediate river discharge threshold that satisfies R2T=1 is expected to vary
with estuarine size and shape, given that the tidal mean discharge is strongly
affected by estuarine morphology. These dynamic complexities merit further
study for site-specific cases.

**5. Conclusions**
Based on past intensive studies of tidal dynamics in estuaries, this work is
devoted to examining the forced overtide behavior under varying river
discharge and the controlling nonlinear mechanism. We use a numerical
model for a schematized long estuary to capture the nonlinear dynamics as
much as possible. Model results reveal local overtide generation whose
amplitude however exhibits strong spatial dependence. While the principal $M_2$
tide is increasingly dissipated as the R2T ratio increases from zero, the $M_4$
overtide amplitude decreases in the upper part of estuaries but increases in
the lower part of estuaries. With increasing R2T ratio, the total energy of $M_4$
overtide integrated throughout the estuary first increases and reaches a peak
when the R2T ratio approaches unit. Further larger river discharge induces a
decline in total energy of both $M_2$ and $M_4$. The modeled nonlinear changes in
overtide are quantitatively validated by data in actual estuaries like the
Changjiang and Amazon River estuaries.
Further sensitivity simulations confirm the significant role of bottom friction
that is enhanced by river-tide interaction in controlling the overtide behavior.
The effective bottom friction is enhanced by the river discharge, and this has
two-fold impact: (1) dissipation of principal tidal energy and (2) stimulation of
energy transfer to overtides. The two-fold effect explains the occurrence of an
intermediate river discharge threshold that benefits maximal overtide
amplitude. This study demonstrates the need to look at both tidal wave
propagation and deformation at the same time in tidal wave dynamics, as well
as their nonlinear spatial variations in large river estuaries. The findings have
implications for study of tidal bores and tidal asymmetry and associated



morphological changes in river estuaries.

**Author contribution**

LG designed the experiments and carried them out. LG and IT prepared the
manuscript with contributions from all co-authors.

**Declare of interest conflict**

The authors declare that they have no conflict of interest.

**Data availability**

The model data are available on request at the corresponding author.

**Acknowledgements**

This work is supported by the project 'Coping with deltas in transition' within
the Programme of Strategic Scientific Alliance between China and The
Netherlands (PSA), financed by the Ministry of Science and Technology, P.R.
China (MOST) (No. 2016YFE0133700) and Royal Netherlands Academy of
Arts and Sciences (KNAW) (No. PSA-SA-E-02), and also partly by MOST (No.
2017YFE0107400), National Natural Science Foundation of China (Nos.
51739005; U2040216; 41876091), Shanghai Committee of Science and
Technology (Nos. 19QA1402900; 20DZ1204700), and the Fundamental
Research Funds for the Central Universities. We thank David A. Jay (PSU,
USA) and Henk M. Schuttelaars (TU Delft, the Netherlands) for their comments
and suggestions.

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
