# Peer review of "River-enhanced non-linear overtide variations in river 1 estuaries 2 3 Leicheng Guo1, \*, Chunyan Zhu1, 2, Huayang Cai3, Zheng Bing Wang1, 2, 4, 4 lan Townend 5, Qing He 1 5 6 7 1 State Key Lab of Estuarine and Coastal"

_Hydrology and Earth System Sciences, 2021_

## Author Comment (AC3)

**RC3**: , Anonymous Referee #2, 27 Apr 2021

This manuscript discusses the effect of river-tide interactions on the generation of overtides, specifically the M4 tide. First, several mechanisms causing the M4 tide in the 1D shallow water equations are computed and analyzed following the method of Gallo and Vinzon (2005). Second, and their main finding, is that the total energy in the generated M4 tide varies with the river discharge and displays a maximum for an, in their range, intermediate discharge. This is further explained conceptually.

I like the idea of the main finding that the energy in the generated M4 tide varies with river discharge and displays a maximum and I think such a thing would be an insightful finding. However, I do not think the conclusions are actually valid and certainly not sufficiently demonstrated. To summarize my main comments (full details are given below):

(1) I have good reasons to think that the conclusions are actually only valid for a few cases that look very much like the chosen case study and carry little generality for other estuaries.

**A:** The reviewer's concerns are noted. However, they are too vague and open ended for us to be able to respond directly to their concerns. In this work, there are two main findings: one is the spatially non-uniform and non-linear changes in overtide amplitude in response to different river discharges, and one is the explanation of the integrated maximal overtide energy using a threshold of R2T=1. The first finding can be partially validated by similar results detected in Amazon (Gallo and Vinzon, 2005), in Columbia Estuary (Jay et al., 2014), Changjiang Estuary (Guo et al., 2015), St. Lawrence Estuary (Matte et al., 2013), and Ganges Delta (Elahi et al., 2020). These case studies only show tidal changes under one or two river discharge conditions, while the modeling work in our study extends this to a range of river discharges from 0 to 90000 $m^3$/s, thus providing a more complete picture. We have also argued in the discussion section why similar phenomenon have not widely reported in many other tide-dominated (small) estuaries given tides are a very basic research theme that have been studied for centuries. "*We think that it maybe because the river discharge in tide-dominated estuaries is comparatively small and*

*rarely reaches a magnitude that exceeds R2T=1. Therefore, the role of river discharge in stimulating tidal wave deformation and overtide generation has been widely observed and confirmed (when R2T<1), whereas further changing behaviors under R2T>1 are far less prevalent and hence less well documented. Another explanation is that most tide-dominated estuaries are relatively shorter in physical length compared with tidal wavelength, hence the distinction between tidal river and tidal estuary and the spatially nonlinear overtide variations are less apparent compared with that in long estuaries with profound river influences.*"

As to the second main finding, it is mainly derived based on the model results in this work, which will inevitably depend on the model settings. However, we note that the findings Elahi et al. (2020), examining the Ganges, are consistent with our finding. Elahi et al. (2020) suggested an optimal balance featured by equal river-induced mean current velocity and tidal-induced current velocity magnitude, whereas we found that R2T=1 when looking at the mouth section, suggesting a level of consistency between the two approaches. To illustrate the validity of this finding, we are modeling more estuaries with varying shape and geometry and will include the results in the revised work as supporting information. Preliminary indications from the additional runs suggest that the results do not change the findings or conclusions of the main text.

**(2) The conclusions about the spatial characteristics of M4 are not supported by the results, which are integrated over the length of the channel.**

**A:** This maybe a misunderstanding. In the Conclusion section, we had stated that '*While the principal $M_2$ tide is increasingly dissipated as the R2T ratio increases from zero, the $M_4$ overtide amplitude decreases in the upper part of estuaries but increases in the lower part of estuaries*'. This statement can be clearly seen from Figure 3b and Figure 4c.

Following that, we concluded that '*With increasing R2T ratio, the total energy of M₄ overtide integrated throughout the estuary first increases and reaches a peak when the R2T ratio approaches unit.*' This latter statement refers to the integrated results, which is clearly supported by Figure 6.

The above two statements are not the same and should not be conflated.

**(3) I think the explanation of the maximum of M4 energy for intermediate discharge as balance between dissipation and generation is incorrect and actually caused by a different mechanism.**

**A:** The reviewer's concerns are noted. However, they are too vague and open ended for us to be able to respond directly to their concerns. It has been widely accepted and understood that 1) the significant overtide $M_4$ in shallow water environments is generated by the nonlinear processes like advection and friction etc.; 2) the energy of $M_4$ derives from its parent tides $M_2$ (Parker, 1991; Wang et al., 1999 etc); and 3) river discharge enhances tidal dissipation of $M_2$ predominantly through the friction effect (Horrevoets et al., 2004). Hence it logically makes sense that the overtide will depend on the generation and dissipation, both of which depends on the nonlinear mechanisms.

**Furthermore, the method employed by the authors is shaky: the key equations that much of the results rely on contain multiple quite essential errors and the case study is very specific (also see details below).**
**This leads me to the recommendation to reject this paper.**
**Main comments about the conclusions**

- **In fact you study the transfer of energy from two harmonic components (subtidal and M2) to another (M4). The generated M4 tide in itself is a wave that may propagate according to its own dynamics. This highlights two big problems with the present analysis:**

**Firstly, your case is friction dominated and for a long estuary without reflection at the head of the estuary. This means that travelling waves will decay. Hence, a small increment in M4 tide generated in some location will not propagate very far. Thus, you dominantly see that locally strong generation of M4 results in a locally strong M4 with some spatial smoothing due to the propagation. I expect that this is totally invalidated in estuaries that are not dominated strongly by friction everywhere or**

**which are shorter and reflect the incoming wave. Hence, your results only represent a small portion of all estuaries. Some of the strongly converging, less frictional branches of the Yangtze estuary itself (which are not considered in this study) could already be a counterexample. Hence, I am of the opinion that a 'general theory' as presented here is not so useful and one could just as well study the handful of actual estuaries satisfying it.**

**A:** It is true that the locally generated $M_4$ tide propagates in the estuary as a wave. However, M4 was not imposed at the seaward boundary in this modeling study. As $M_4$ is generated inside the estuary, it propagates, but also continues to be generated by the transfer of energy from higher harmonics further upstream. Hence the $M_4$ amplitude does not necessarily reduce in the landward direction in the same way as $M_2$. Both model results and observed results in many actual estuaries confirm the spatial variations of the $M_4$ tide, first increasing from the river mouth and then decreasing in the landward direction, e.g., in the Amazon and Changjiang estuaries. We do not expect anything else than this behavior, as at the mouth the $M_4$ was not imposed at the boundary. The amplitude of $M_4$ tide is not merely determined by the local generation and we do not agree with reviewer's statement "locally strong generation of $M_4$ results in a locally strong $M_4$ with some spatial smoothing due to the propgation". The local generation will for sure also influence the $M_4$ amplitude further landwards.

We see that in some very special cases, e.g. estuaries that are not dominated by friction, are shorter, or where wave reflection is important, as mentioned by the Reviewer, different patterns may indeed emerge. However, we make no claim that our results apply to these cases. We simply note that friction has been shown to be one of the most important mechanisms in damping incoming astronomical tides in many real-word estuaries. The overtide generation still occurs in a frictionless environment, but its amplitude becomes smaller and the nonlinear behavior of along-channel change in response to increasing river discharge does not show up (see Figure S4 in the Supporting Information). Wave reflection is not expected to be large in

estuaries with high river discharge which will damp the incoming tides to a large degree. This study and the associated findings and statements should be applicable to large river estuaries with considerable river discharge variations, but maybe not to some special systems, such as a barrier dam within tidal wave limit that causes wave reflection etc.

**Secondly, you explain the maximum in integrated M4 energy for river discharge as a balance between dissipation of the river flow on the M2 tide vs generation of M4 by tide-river interaction (Fig 8). This is not necessarily true. What you actually find is redistribution from the subtidal and M2 water motion to other frequencies. This happens primarily through the term u|u| in the bottom friction, which you may easily show has a maximum for (approximately) R2T=1. So the actual generation of M4 has a maximum. Dissipation is an additional effect but I would guess it is not essential.**

**A:** There maybe another misunderstanding. When we talked about dissipation, we mainly referred to the dissipation of $M_2$, not dissipation of $M_4$. As mentioned above, the $M_4$ is generated by the transfer of energy from $M_2$, the parent tide of $M_4$. So $M_4$ generation inside estuaries is one the dominant processes. Additionally, $M_2$ was predominantly dissipated inside estuaries owing to friction and river discharge, hence dissipation of the $M_2$ tidal energy in the landward direction will affect the energy available to $M_4$, and then affect the generation of $M_4$. This balance of $M_2$ dissipation and $M_4$ generation then controls the net amplitude of $M_4$.

**In section 4.1 and figure 7 you then draw some of the main conclusions on how the local R2T affects the local M4 generation. This is not addressed by your theory, which considers the total integrated M4 energy. Therefore this conclusion is not supported by your results. I expect that this conclusion indeed works in the friction dominated – long estuary setting here but not in general, where the M4 may propagate.**

**A:** The local $M_4$ amplitude is indeed controlled by both the local generation and the propagation of its component generated in the more seaward domain. However, we did not 'draw conclusion on how the local R2T affect the local $M_4$ generation'. As can be seen in response to previous questions, in this study we

firstly detected nonlinear variations of $M_4$ amplitude along the estuary under different river discharge, which depends on the local M4 generation and large-scale propagation. Following that, we integrated the $M_4$ energy throughout the estuary and found a maximal value when scaled with the R2T at the mouth section. Hence, the threshold R2T=1 is not a local influence, but represents the overall impact of river on tides and an intermediate balance between the strength of river and tidal forcing. The findings in this study do indeed mainly apply to long estuaries with friction as an important controlling mechanism in tidal wave propagation. We have clarified that in the discussion to get rid of such confusion, e.g., at the end of section 4.1, we added that '*Hence, the findings in this work mainly apply to long estuaries with significant river discharge and friction control on tidal propagation in which wave reflection is limited.* '

**Ln 491-492 actually address the phase of the M4 (relative to the M2). You don't show any results related to the phase, so this conclusion cannot be drawn.**

**A:** The sentence in line 491-492 was '*These field data and model results confirm that the findings regarding the spatial dependence of overtide on river discharge are likely to be ubiquitous for river estuaries.*' It said nothing about tidal phase. We do not address the phase of $M_4$. To get rid of any possible confusion, we rephrased the sentence as '*These field data and model results confirm that the findings regarding the spatial dependence of overtide amplitude on river discharge are likely to be ubiquitous for river estuaries.*'

**In 554-556: 'In this work we see that the quadratic bottom stress term also leads to significant M4, through river-tide interaction': this is stated as the main novelty, but is not new.**

**A:** The whole sentence mentioned here was '*In this work we see that the quadratic bottom stress term also leads to significant $M_4$, through river-tide interaction, i.e., between a river-enhanced mean current and $M_2$ current.*' We did not emphasize that this is the main novelty. To get rid of any potential

confusion, the sentence is rephrased as '*Additionally, the quadratic bottom stress term also leads to significant $M_4$, through river-tide interaction, i.e., between a river-enhanced mean current and $M_2$ current (Wang et al., 1999).*'

**Section 3.2: I don't see the hypothesis underlying this section. Your case without convergence still features a friction-dominated M2 tide. Since the M2 is similar to the case with convergence, I don't see why the M4 generation should be so different. In any case, just one example of a case without convergence does not prove much. This section does not add anything for me.**

**A:** Section 3.2 was dedicated to showing the possible impact of width convergence on the tidal changes. After comparing the results in Figures 3 (convergent estuary) and 4 (rectangular estuary), we see that the longitudinal changing behaviors of both $M_2$ and $M_4$ are similar, other than the firstly amplified $M_2$ and larger $M_4$ amplitude in the convergent estuary. The incoming $M_2$ tide was predominantly damped inside the rectangular estuary. The latter differences are understandable because width convergence enhances tidal amplification. Given the spatial changing behaviors are the same for both the convergent and non-convergent estuaries, the discussion in the sections following Section 3.2 are based on the results in the non-convergent estuary, in order to isolate the impact of width variations and consequent tidal wave amplification. Overall we think the message in Section 3.2 is helpful to indicate the sensitivity of the overtide changes to different shaped estuaries.

**Main comments about the method**

**Equations (3) and (4) are inconsistent. You assume that only a subtidal and M2 water motion are present, but the numerical computation also allows for all overtides. Implicitly, you assume here that all overtides are much smaller than subtidal and M2, i.e you employ scaling (you do this explicitly on ln 283). This is weird, because in ln 184-195 you argued why models based on scaling analyses are not good enough for your study and you need to use a fully numerical model. If you want to do this, I'd recommend using scaling analysis formally in the analysis. This becomes problematic when the M4 tide is not small compared to M2.**

**A:** The reviewer did not point out and we do not see why Eq. 3 and 4 are inconsistent. In lines 184-195, the point being made is that analytical solutions of governing tidal dynamic equations may not fully capture the nonlinear dynamics, which are important for overtide generation. The analytical solutions adopted scaling analysis in order to simplify the governing equations, e.g., linearize the friction term. The scaling analysis mentioned here is not to evaluate the relative importance of $M_4$ to $M_2$. In this work we used a numeric model which includes all the nonlinear processes represented in the governing equations and can thereby reproduce their non-uniform spatial changes under different river-tidal conditions. But we accept that both analytical and numerical approaches have their own advantages.

In line 283, the whole sentence was '*The bottom friction term is approximately expanded into a bottom shear stress term and a term considering depth variations, as the two terms on the right hand of Eq. (7), respectively, according to Godin and Martinez (1994), given the tidal amplitude to water depth ratio ($|\eta|/h$) is generally smaller than one*', so the scaling was used to show that the tidal wave amplitude is relatively smaller with respect to the water depth, so the bottom friction can be expanded as that shown in Eq.7 according to Godin and Martinez (1994). Given $M_2$ is the main tidal constituent, adding another $M_4$ will not fundamentally change the tidal amplitude to water depth ratio ($|\eta|/h$).

The Eq. 3 and 4 did not include the $M_4$ component because they were used to represent the condition at the river mouth. In this study only an astronomical $M_2$ tidal constituent was prescribed at the seaside boundary and $M_4$ tides are generated inside the estuary. Moreover, including the $M_4$ component in Eq. 3 and 4 will not fundamentally change the results in Eq. 5-7 because $M_2$ amplitude is a factor larger than that of $M_4$. To get rid of the confusion, we have added a sentence in the revision '*Note that the internally generated $M_4$ component is so far not included and shown in Eq. 3 and 4, but*

*this does not fundamentally change the results, given that the amplitude of $M_2$ is much larger than $M_4$.'*

**Eq 10 and therefore 11-13 (i.e. the main decomposition that you rely on in the results) is wrong. This is not what Godin (1999) uses. You need to use a scaling factor U here:**
**u|u| = U(a\*u_scaled + b\*u_scaled^3),**
**such that u_scaled ranges between -1 and 1. On a more detailed level, the coefficient a, b you choose in Eq. 10 are Heron's approximation while Godin (1999) argues that one should better use Chebyshev's approximation.**
**Eq 11-13 contains another mistake: 'theta' is forgotten everywhere. Hence the phase information is lost. This is essential.**
**I am not entirely convinced of the comparison (fig 5) between the 'discharge gradient' term (Eq 11) to the advection and friction terms (Eq. 12-13). The first appears in the continuity equation and the latter terms in the momentum equation. To create the same unit for all terms, you scale with two different quantities, but why can I compare these? I know Gallo & Vinzon (2005) did the same, but to me this is a very inexact analysis. I think you may at most compare the results on order of magnitude and conclude that all terms are of a similar order of magnitude.**

**A:** The above questions focus on the decomposition method and the associated results in Figure 5. We have given more thoughts to the methods to compare the contribution of different nonlinear terms (Eq. 11-13) and we noticed that the issue of different scaling may undermine the quality of the analysis. Given the results in Figure 5 are largely in line with previous studies- showing that the friction term dominates the overtide change behavior- this does not add much new understanding. In this revision we have therefore removed the discussion on the contribution of the nonlinear terms (section 3.3 and Figure 5) and associated texts on the methods to get rid of confusion. The main findings of this work are then 1) highlighting the spatially nonlinear changes of overtide amplitude under varying river discharges and 2) identification of the threshold condition controlling the maximal overtide generation.

**Other remarks**

**Ln 184-191: I don't think this gives a proper reflection of the literature. Some of the analytical or semi-analytical literature actually resolves (part of the) overtide and various nonlinear terms, e.g. Friedrichs & Aubrey (1988), Lanzoni & Seminara (1998), Ridderinkhof et al (2014), Alebregtse & de Swart (2016), Chernetsky et al (2010), Dijkstra et al (2017). Indeed full treatment of the nonlinearities is not done this way, but since the M4 tide is still generally small compared to the M2 tide, these methods could still work.**

**A:** Thank you for the suggestion. We have revised the sentences into '*Analytical models usually assume tidal propagation as a single wave component, based on simplified tidal dynamic equations after scaling analyses, e.g., adopting a linear assumption or a nonlinear expansion of the friction term (Green, 1837; Kreiss, 1957; Jay, 1991; Parker, 1991; Friedrichs and Aubrey, 1994; van Rijn, 2011). Further improved analysis method can take into account of more than one tidal component and robust approximation of the nonlinear friction term and their impact on morphodynamic changes (Lanzoni and Seminara, 1998; Ridderinkhof et al., 2014; Alebregtse and de Swart, 2016; Chernetsky et al., 2010; Dijkstra et al., 2017).*' The suggested references are included in the reference.

**Ln 226-241: I don't think a morphodynamic computation is necessary at all. One could just compute hydrodynamics for a given bathymetry (this would be different when computing sedimentation rates or such). If you do this: what is the final bathymetry?**

**A:** Indeed, the tidal hydrodynamics within an estuary can adapt to any arbitrary bathymetry. However, this may only be stable if the bed is erodible. That is why, in this work we first run a morphodynamic simulation to obtain a close-to-equilibrium bed profile for the defined tidal forcing conditions and then use the equilibrium bed profile in further tidal simulations. We believe this treatment is preferred to minimize the impact of an arbitrary bathymetry on the tidal dynamics, thus highlighting the effect of river discharge. Even though, our previous study (Guo et al., 2016) did show that the overall overtide changes in response to river discharge variations exhibit similar spatial pattern under initial and equilibrium bed profiles (see Figure R1 below).

[Figure]

Figure R1. Longitudinal changes of $M_2$ and $M_4$ amplitude modeled in a 560-km long rectangular estuary forced by different river discharge under primary initial bed profile (dotted lines) and equilibrium bed profiles (solid lines) (from Guo et al., 2016)

**Ln 263-264: how can the depth be constant after the morphodynamic computation?**

**A:** It is a mistake in using the word 'constant'. What we mean was that the water depth will not change over time anymore. So the sentence is corrected as '*As the bed level is prescribed as an equilibrium profile, the water depth h remains unchanged*'.

**Eq 8: brackets missing in the cosine.**

**A:** A bracket is added.

**Eq 13: is a minus missing in the first term or did I get confused with the sign of u0?**

**A:** The minus sign depends on the reference direction of current. A minus sign for $u_0$ is needed if assuming flooding currents are defined positive. So a minus sign was added in Eq. 4. The Eqs. 11-13 were removed in this revision to get rid of confusion.

**Ln 332: why do you need the M4 amplitude and phase? It does not appear in Eq. 11-13.**

**A:** Both $M_2$ and $M_4$ amplitude and phases are output of the harmonic analysis, but only $M_2$ harmonics are used in Eq. 11-13.

**Section 3.1: I missed the calibration or setting of friction parameter. How was this done?**

**A:** As a schematized model was employed in this study, we did not do calibration of the friction parameter against field data. In the section 2.1, we added that '*A uniform friction coefficient of Chézy value of 65 $m^{1/2}$/s was used, which leads to predominantly landward tidal damping within the schematized estuary, comparable to what is observed in reality (see Figures 1 and 3)*.'

**Ln 360-363: why include S2 now? This seems inconsistent with the entire method section.**

**A:** Yes, the major part of this study focused on the $M_2$ and $M_4$ changes. The $S_2$ tide was mentioned here to show that other compound tide like $MS_4$, which was generated by $M_2$-$S_2$ interaction, exhibits similar changes as $M_4$ in response to river discharge (Figure 3c).

**Fig 5a: you find a contribution from bottom friction while there is no discharge. Is this the effect of tidal return flow?**

**A:** Yes, the $u_0$ velocity is nonzero in the no river discharge scenario, owing to a return flow under the predominantly progressive wave condition.

**Ln 495 'majority of estuaries': I don't think this is obviously true. This would at least need a reference.**

**A:** The sentence was rephrased and references are included to get rid of confusion. In the revised work, the new sentence is '*Note that the above-mentioned nonlinear overtide changes were predominantly reported in large, mixed river-tide energy estuaries and deltas, e.g., Amazon (Gallo and Vinzon, 2005), Changjiang (Guo et al., 2015), and Ganges (Elahi et al., 2020), but less in many other tide-dominated estuaries with relatively smaller river discharge, although the importance of overtide in controlling tidal asymmetry*

*and residual transport were widely reported, e.g., in Humber in the UK (Winterwerp, 2004) and western Scheldt Estuary in the Netherlands (Wang et al., 2002).'*

**Ln 424-435: why discuss this here. You don't seem to do this explicitly, so this is more a discussion to me. I don't find this very insightful, because naturally the linearized friction does not contain any transfer of energy from one frequency to another.**

**A:** We aimed to stress the role of nonlinearity of the friction in controlling river-tide interaction, thus a comparison of the model results between the nonlinear (default) and linear friction was conducted. As the discussion of the different contribution of the nonlinear terms (original section 3.3) was removed in this section, the texts on the discussion of the sensitivity to linear friction have been moved into section 4.2 as part of the discussion instead of result.

**Ln 525-527: you should either prove this or don't mention it.**

**A:** The changing behavior of $MS_4$ was shown in Figure 3c. The sentence was removed to be more focused on overtide $M_4$.

**Ln 531-535: I can't follow this. Again this refers to the local discussion I commented on earlier.**

**A:** The sentence '*In the lower part of estuaries where the incident tides are less dissipated, river flow plays a more important role in reinforcing the effective bottom friction. As a result, dissipation of the principal tide is more prominent in the upper part of estuaries, while tidal energy transfer and overtide generation are more substantial in the lower part of estuaries (Figures 8)'* was to stress the different impact of river flow on the tidal changes in seaward and landward part of an estuary. As we had argued in the response to previous questions, the spatial variations of overtide $M_4$ along the estuary are a combined result of dissipation of $M_2$ and generation of $M_4$. To further clarify this point, the sentence is rephrased in this revision as '*In the seaward part of estuary where the incident tidal waves are less dissipated by increasing river*

*discharge, the role of the river discharge in reinforcing the effective bottom friction and enhancing overtide generation is more pronounced. Whereas, in the landward part of the estuary, river discharge plays a prominent role in causing more dissipation of the principal tide. As a result, the tidal energy available to be transferred to overtide is constrained, thus overtide amplitude reduces with increasing river discharge in the landward regions (Figures 8)'.*